


# Characterization of the OCO-2 instrument line shape functions using on-orbit solar measurements

Kang Sun[1], Xiong Liu[1], Caroline R. Nowlan[1], Zhaonan Cai[1], Kelly Chance[1], Christian Frankenberg[2,3], Richard A. M. Lee[3], Randy Pollock[3], Robert Rosenberg[3], and David Crisp[3]

[1]Harvard-Smithsonian Center for Astrophysics, 60 Garden Street, Cambridge, MA, USA
[2]Division of Geological and Planetary Science, California Institute of Technology, Pasadena, USA
[3]Jet Propulsion Laboratory, California Institute of Technology, Pasadena, CA, USA

*Correspondence to:* Kang Sun (kang.sun@cfa.harvard.edu)

**Abstract.** Accurately characterizing the instrument line shape (ILS) of the Orbiting Carbon Observatory-2 (OCO-2) is challenging and highly important due to its high spectral resolution and requirement for retrieval accuracy (0.25 %) compared to previous space-borne grating spectrometers. On-orbit ILS functions for all three bands of the OCO-2 instrument have been derived using its frequent solar measurements and high-resolution solar reference spectra. The solar reference spectrum gen-

erated from the 2016 version of the TCCON solar line list shows significant improvements in the fitting residual compared to the solar reference spectrum currently used in the v7 L2 algorithm in the $O_2$ *A* band. The analytical functions used to represent the ILS of previous grating spectrometers are found to be inadequate for the OCO-2 ILS. Particularly, the hybrid Gaussian and Super Gaussian functions may introduce spurious variations, up to 5 % of the ILS width, depending on the spectral sampling position, when there is a spectral undersampling. Fitting a homogeneous stretch of the preflight ILS together with the relative

widening of the wings of the ILS is insensitive to the sampling grid position and accurately captures the variation of ILS in the $O_2$ *A* band between decontamination events. These temporal changes of ILS may explain the spurious signals observed in the solar-induce fluorescence retrieval in barren areas.

## 1 Introduction

The Orbiting Carbon Observatory-2 (OCO-2), launched on 2 July 2014, is a NASA mission aiming at quantifying the sources

and sinks of $CO_2$ at regional scales (100–1000 km) (Crisp, 2015). OCO-2 will also be able to characterize the global $CO_2$ seasonal cycles and annual variations. To achieve its mission goal, OCO-2 was designed with significantly higher precision, spectral and spatial resolution, and spatial coverage requirements than existing satellite instruments. The OCO-2 instrument aims to measure the column-averaged $CO_2$ dry air mole fraction, $X_{CO_2}$, with uncertainties near 1 ppmv (0.25 % of current $X_{CO_2}$) on regional-to-continental scales (Crisp et al., 2004; Crisp, 2008; Frankenberg et al., 2015). The OCO-2 instrument

incorporates three imaging grating spectrometers optimized for narrow spectral ranges around 765 nm ($O_2$ *A* band, or O2A herein), 1.61 $\mu$m (weak $CO_2$ band, WCO2), and 2.06 $\mu$m (strong $CO_2$ band, SCO2) with a resolving power ($\lambda/\Delta\lambda$) of $\sim$ 20,000 (Eldering et al., 2015). The O2A band absorption directly constrains the dry air column abundance and the atmospheric optical path length. The WCO2 and SCO2 bands provide information about both the $CO_2$ column abundance and aerosol





properties. Each spectrometer produces spectra for eight spatial footprints with 1016 spectral pixels (Eldering et al., 2015). The small footprint size ($< 1.3 \times 2.3$ km$^2$ at nadir) helps to minimize the impact of clouds and facilitates the detection and quantification of the emissions by small-scale sources, e.g., power plants and cities (Crisp, 2015).

In order to retrieve $X_{CO_2}$ with 1 ppmv uncertainty, the instrument line shape (ILS) must be accurately determined. The need to characterize variations in the ILS across the 1016-pixel spectral range for each of the 8 footprints in the 3 spectrometer channels poses a central challenge to the OCO-2 spectral calibration. Currently, the OCO-2 retrieval algorithm uses the preflight measured ILS tabulated as look-up tables (Day et al., 2011; Frankenberg et al., 2015; Lee et al., 2016). However, the vibration during launch and the thermal, gravitational, and radiative contrasts between the space and laboratory conditions may introduce subtle changes in the ILS. The on-orbit thermal variation, instrument degradation, and switching of observation modes may also cause ILS variations, as observed by other satellite instruments (De Smedt et al., 2012; Miles et al., 2015). Therefore, it is necessary to characterize the on-orbit behavior of the ILS throughout the mission.

On-orbit ILS and wavelength registration calibrations for existing grating spectrometers (GOME, GOME-2, OMI, etc.) are typically performed by fitting the measured solar irradiance at the top of the atmosphere with a well-calibrated, high-resolution solar irradiance reference spectrum, assuming analytical function forms of ILS (Chance, 1998; Liu et al., 2005, 2010; Cai et al., 2012; De Smedt et al., 2012; Munro et al., 2016). Compared to these space-borne instruments that generally targeted the UV/visible bands, OCO-2 resolves rotational vibrational bands of O$_2$ and CO$_2$ in the infrared at much finer resolution over a narrower spectral range (e.g., the spectral resolution of OCO-2 is one order of magnitude higher than GOME, GOME-2, and SCHIAMACHY in the O2A band). As a result, a solar reference spectrum with even higher resolution is necessary. The solar lines are significantly weaker, and there are fewer solar lines in the infrared than in UV/visible bands, which introduces additional challenges. The retrieval accuracy requirement for OCO-2 ($\sim 0.25$ % for $X_{CO_2}$) is also much higher than that for the species measured by existing satellite instruments, so small ILS differences that may be tolerated in other instruments may jeopardize the $X_{CO_2}$ retrieval. To this end, the aim of this study is to perform on-orbit OCO-2 ILS calibrations using the instrument's frequent solar irradiance measurements and evaluate the temporal and inter-footprint variation of ILS functions during the mission.

## 2 Instrument and data analysis

### 2.1 OCO-2 instrument and its solar measurements

OCO-2 is based on the original OCO mission (Crisp, 2008) that did not achieve orbit due to a failure of the launch vehicle. The OCO-2 instrument and mission have been described in detail by Crisp (2015). The three spectrometers targeting O2A, WCO2, and SCO2 bands use similar optical designs and are integrated into a common structure to improve system rigidity and thermal stability. They are fed by a common F/1.8 Cassegrain telescope through a series of beam splitters and re-imagers. Each spectrometer produces an image on a $1024 \times 1024$ pixel focal plane array (FPA). In the spectral direction, 1016 out of 1024 pixels are used. The typical full widths at half maximum (FWHM) of ILS are around 0.04, 0.08, and 0.1 nm for the O2A, WCO2, and SCO2 bands, respectively. Each FWHM is sampled by 2–3 spectral pixels. In the spatial direction, only





$\sim 190$ out of 1024 pixels receive photons, limited by the slit length, and the science measurements are restricted to the middle $\sim 160$ pixels. For routine science observations, this $160 \times 1016$ pixel "frame" is recorded 3 times each second. The 160 spatial pixels are then summed into 8, $\sim 20$-pixel bins or "footprints". Malfunctioning pixels, defined by a bad pixel map, are excluded during the summation (Crisp et al., 2016; Rosenberg et al., 2016).

The OCO-2 instrument observes the Sun through a transmissive diffuser to reduce the incident irradiance. Note that the solar diffuser does subtly change the ILS due to its uneven illumination of the telescope aperture (see Crisp et al. (2016) for images of the diffuser). Routine observations of the solar spectra are conducted near the northern terminator shortly after final science measurements for a given orbit. A regular solar observation lasts for about 1 min, yielding $\sim 180$ frames of solar spectra. About once per month or after each instrument decontamination (decon), a special solar Doppler measurement is performed,

where solar observations are collected during the entire dayside of an orbit. About 11,000 frames of solar spectra are collected continuously from near the south pole to near the north pole. Figure 1 shows the ranges of relative radial velocity of the spacecraft to the Sun during solar observations. The regular solar measurements are carried out at $\sim 7$ km s$^{-1}$ relative velocity to the Sun (red shift), whereas the Solar Doppler measurements span from $-7$ to 7 km s$^{-1}$ (blue to red shift).

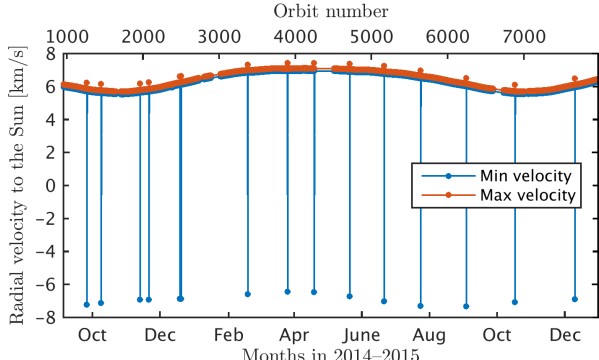

**Figure 1.** Ranges of relative radial velocity of the OCO-2 spacecraft to the Sun during solar observations. The occasional wide ranges correspond to solar Doppler measurements.

For each regular solar measurement, the $\sim 180$ solar frames are averaged with the lowest and highest 5 % irradiance values

trimmed. This trimmed averaging helps to remove cosmic ray contamination that sometimes causes positive anomalies up to 20 times higher than the solar irradiance. One example of a regular solar measurement by all three bands is shown in Fig. 2. For solar Doppler measurements, all frames are binned into 100 intervals according to the relative velocity to the Sun. The frames within each interval are then similarly trimmed and averaged, yielding 100 solar spectra at varying degrees of blue/red shift. These solar spectra are then corrected for the Doppler shift, merging them into one single, highly oversampled solar spectrum.

The solar Doppler measurement ($-7$ to 7 km s$^{-1}$) moves the spectrum by over two times the spectral sampling interval. In the following analyses, only the fraction from $-3$ to 3 km s$^{-1}$ is used to construct the oversampled solar Doppler spectra to avoid earthshine contamination sometimes seen at high latitudes.





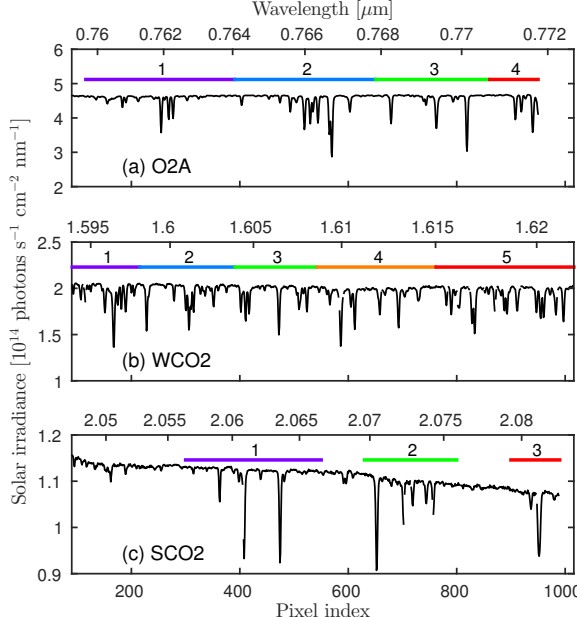

**Figure 2.** OCO-2 solar spectra measured at O2A (a), WCO2 (b), and SCO2 (c) bands. The colored horizontal bars indicate the spectral windows within which ILS functions are fitted. The solar spectra are from footprint 1, orbit 3928 on 29 March 2015.

The OCO-2 ILS as well as wavelength registration of each spectral pixel were measured by stepping a tunable diode laser through the OCO-2 spectral range before launch (Day et al., 2011; Lee et al., 2016). The final ILS and wavelength registration were then optimized and validated by comparing atmospheric observations to simultaneous observations from a co-located TC-CON station (Frankenberg et al., 2015). Various combinations of conventional analytical line shape functions (Gaussian, Voigt,

5    Lorentzian, etc.) were tested to fit the measured ILS but could not match the telluric spectra well enough for accurate $X_{CO_2}$ re-trievals. Therefore, these preflight ILS results were digitized and saved as a $3 \times 8 \times 1016 \times 200$-element look-up table. Namely, the ILS for each band, footprint, and spectral pixel is defined at 200 spectral points around the center point (Lee et al., 2016). Figure 3a shows an example of the tabulated preflight ILS. The wavelength registration is expressed by fifth-order polynomial spectral dispersion coefficients that map spectral pixel index to wavelength. The preflight ILS and wavelength calibration have

10   been used in the operational $X_{CO_2}$ retrieval algorithm. Wavelength shift/squeeze terms are retrieved for each sounding in the Level 2 (L2) algorithm, but the ILS have been assumed to be constant. Both static spectral dispersion coefficients and preflight ILS are provided in OCO-2 Level 1B data (OCO-2 Science Team et al., 2015).



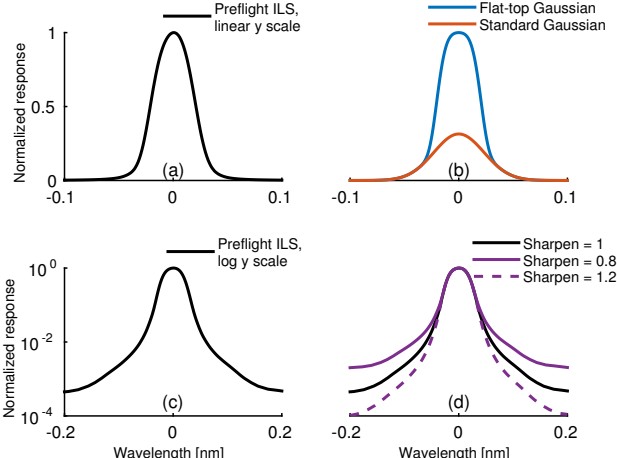

**Figure 3.** (a) shows an example of the tabulated preflight ILS at 766 nm for footprint 5. (b) shows the derived hybrid asymmetric Gaussian, decomposed into flat-top and standard Gaussians, from fitting the corresponding solar spectrum microwindow (window 2 at the O2A band, see Fig. 2). (c) is the same as (a) but in log y scale. (d) shows the differences between the sharpen terms of 0.8, 1, and 1.2. The stretch term is kept at unity.

## 2.2 ILS fitting algorithm

The OCO-2 solar spectra ($I_0(\lambda)$) can be modeled by convolving a high resolution solar reference spectrum with assumed ILS functions:

$$I_0(\lambda) = \frac{\int_{\lambda'} I_{h0}(\lambda') S(\lambda + \delta\lambda - \lambda') d\lambda'}{\int_{\lambda'} S(\lambda') d\lambda'} \times \sum_{i=0}^{m} P_i (\lambda - \bar{\lambda})^i, \tag{1}$$

where $I_{h0}$ denotes the high resolution solar reference spectrum, $S(\lambda)$ denotes the ILS function, $\lambda + \delta\lambda$ indicates the wavelength calibration/registration (e.g, shift and squeeze, used in this study, or polynomial), and $P_i$ are the scaling $m$th-order polynomial coefficients. The solar reference spectrum is comprised of a pseudo-transmittance spectrum and a solar continuum spectrum. The pseudo-transmittance spectrum is generated from an empirical solar line list developed for the TCCON project (Toon, 2014). This solar line list has been derived by simultaneous fitting of multiple high resolution ground-based, air-, and space-

borne Fourier transform spectrometer (FTS) solar spectra. The current OCO-2 L2 retrieval algorithm (version 7, v7) also uses the 2013 version of the solar line list to generate solar absorption lines in its solar model. The line list was updated in 2015 and 2016 with significant improvements, especially in the O2A band (Toon et al., 2015). The solar pseudo-transmittance spectra generated at 0.01 cm$^{-1}$ from these line lists are available from http://mark4sun.jpl.nasa.gov/toon/solar/solar_spectrum.html. This resolution is more than 10 times higher than OCO-2 spectral sampling intervals and sufficient to resolve the ILS. The 2013,

2015, and 2016 versions of solar line lists are compared in this study. The solar continuum is adopted from the OCO-2 L2 solar model, which was originally derived from the low-resolution extra-terrestrial solar spectrum acquired by the Solar Spectrum





(SOLSPEC) instrument (Thuillier et al., 2003; Boesch et al., 2015). The product of the pseudo-transmittance spectrum and solar continuum gives high-resolution, absolutely-calibrated solar reference spectrum.

For previous GOME, GOME-2, and OMI satellite instruments, the ILS function, $S(\lambda)$, has been modeled by standard or modified Gaussian functions (Chance, 1998; Liu et al., 2005; Dirksen et al., 2006; Liu et al., 2010; Cai et al., 2012; De Smedt

et al., 2012; Munro et al., 2016). The preflight OCO-2 ILS at O2A and WCO2 bands show a significantly broader top compared to a Gaussian shape (Frankenberg et al., 2015). Therefore, a broadened Gaussian shape slit function is implemented in this study as hybrid combination of an asymmetric standard and an asymmetric flat-top Gaussian function (hereafter referred to as hybrid asymmetric, see Fig. 3b for an illustration of its shape), similar to Liu et al. (2015) and Nowlan et al. (2016):

$$S(\Delta\lambda) = (1-w)\exp\left[-\left(\frac{\Delta\lambda}{h_g(1+\mathrm{sgn}(\Delta\lambda)a_g)}\right)^2\right] + w\exp\left[-\left(\frac{\Delta\lambda}{h_t(1+\mathrm{sgn}(\Delta\lambda)a_t)}\right)^4\right], \qquad (2)$$

where $w$ is the relative weighting between the standard and flat-top Gaussian, $h_g$ and $h_t$ are half-width at $1/e$ for standard and flat-top Gaussian, $a_g$ and $a_t$ are their asymmetries, and sgn() denotes the sign function. These five parameters are fitted simultaneously using Eq. (1) by a non-linear least square fitting routine. However, the preflight ILS measurements also showed significant spectral variations; the SCO2 band as well as the low wavelength ends of the O2A and WCO2 bands are closer to Gaussian. To study the impact of using different analytical functions, hybrid symmetric (fixing $a_g$ and $a_t$ at zero) and Gaussian

asymmetric (fixing $w$ at zero) ILS functions are also fitted using the same routine.

The preflight OCO-2 ILS functions show significantly irregular fine structures in the wings (see Fig. 3c), which cannot be fully represented by analytical functions (Frankenberg et al., 2015). The full physics L2 retrieval tests also showed that the preflight ILS produced better retrievals with Earth spectra than analytical functions. Hence, two other ILS functional forms are fitted to preserve the fine structures of preflight ILS and only adjust the general shape. One is similar to Day et al. (2011), where

the preflight ILS is scaled in the $\Delta\lambda$ axis (stretch term), and then the entire ILS is raised to a certain power (sharpen term) but then rescaled in the $\Delta\lambda$ axis to keep the FWHM unchanged. Hence the FWHM is only controlled by the stretch term, whereas the sharpen term determines the shape. Figure 3d shows the effect of changing the sharpen term, which broadens/compresses the wings of the ILS. The other ILS fitting adjusts the stretch term only. These two are referred to as "stretch/sharpen" and "stretch only" hereafter.

Only a limited number of solar lines are covered by OCO-2 spectral windows, especially at O2A and SCO2 bands, and the solar lines are relatively shallow (Fig. 2). To optimize the sensitivity to ILS and to minimize the computational time, each band is firstly divided into 3–5 windows containing strong solar lines (marked as colored horizontal bars in Fig. 2), and a single ILS function is fitted within each window. The window boundaries are chosen at solar continuum regions so that wavelength shift, which is less than 1–2 pixels, will not significantly change solar features covered by the window. Sliding windows with various

sizes and increments as used by Cai et al. (2012) and Liu et al. (2015) were tested, but did not give smooth ILS variations due to sparse solar lines. For the "stretch/sharpen" and "stretch only" fittings, the preflight ILS at the median spectral pixel is applied to the entire window.



One caveat is that these derived ILS functions from solar spectra would show the combined changes due to the instrument and the solar diffuser. Hence the results will be validated by looking at spectra with the solar diffuser out of the way (Section 7).

## 3  Wavelength calibration of solar spectra

The wavelength shift and squeeze of solar spectra are fitted by applying Eq. (1) over the entire band, assuming one of the ILS functions listed above (hybrid asymmetric, hybrid symmetric, Gaussian asymmetric, "stretch/sharpen", and "stretch only"). The wavelength shift fitting results are very similar using the symmetric ILS functions (hybrid symmetric, "stretch/sharpen", and "stretch only") with differences less than 1 % of the derived wavelength shift, or $\sim 10^{-4}$ nm. The wavelength shift results using asymmetric ILS functions (hybrid asymmetric and Gaussian asymmetric) show larger random errors than the other ILS
forms, up to 10 % of the derived wavelength shift, because of the competing effects of the asymmetry term(s) and wavelength shift.

Figure 4 shows wavelength shift and squeeze terms for the regular solar measurements derived using the "stretch only" fitting. The wavelengths always have a red shift corresponding to $\sim 7$ km s$^{-1}$ from the Sun. The annual cycle of wavelength red shift, consistent for all three bands (Fig. 4a, c, and e), can be directly explained by the annually varying velocity from
the Sun (Fig. 1). Between November 2014 and July 2015, the wavelength shift shows stepwise changes corresponding to the switching of nadir/glint observation modes on alternate 16-day global ground-track repeat cycles. After July 2015 when OCO-2 modified its observing strategy so that nadir/glint orbits are interwoven, the glint/nadir differences became less significant. This stepwise wavelength shifts are caused by the different equilibrium thermal states during nadir and glint modes. The thermal gradients across the optical bench assembly shift the image on the FPA by a few microns, leading to wavelength shift difference
between nadir/glint modes.

The squeeze term (Fig. 4b, d, and f) does not show clear annual cycles or correspondence to the observing mode. It is mainly influenced by the cryo-cooler that cools all three FPAs to $\sim 120$ K. During the decon processes, the cryo-cooler is turned off, and the FPA heaters raise the temperature to $\sim 300$ K. Following the decon, ice accumulates on the thermal straps connecting the cold-head and FPAs. This changes the thermal emissivity of the straps and, over time, causes the cryo-cooler to work harder
to provide the same cooling power. This, in turn, resulted in a slight mechanical tilt of the FPAs, relative to the optical axis, which leads to the corresponding slow change observed in the squeeze term (Crisp et al., 2016). This same effect is observed in the dispersion stretch term of the Earthshine spectra, retrieved in the L2 full physics algorithm. The fitted squeeze term for SCO2 band is more noisy due to sparse solar lines and therefore weaker constraints on the wavelength stretch/squeeze. The shift and squeeze terms are different for the eight spatial footprints, most noticeably in the shift term of SCO2 band (Fig. 4e)
and the squeeze term of WCO2 band (Fig. 4d), indicating that the spectral dispersion coefficients have changed differently between preflight and on-orbit for each spatial footprint.





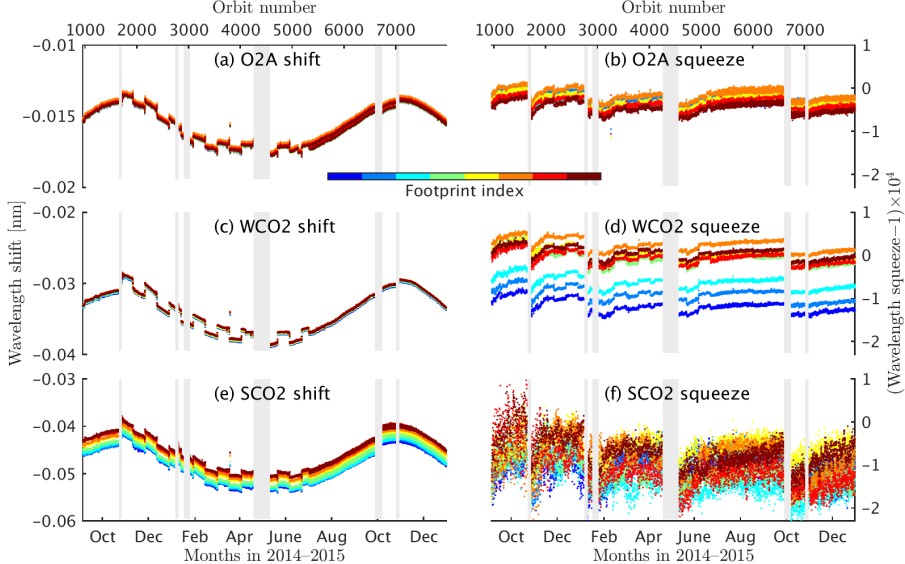

**Figure 4.** Temporal variation of wavelength shift and squeeze terms of the measured solar spectra for all spatial footprints of all OCO-2 bands. The stepwise features in the wavelength shift (a, c, e) are due to transitions between nadir and glint observations which lead to different instrument temperatures. The gaps of data (marked as gray bands) in all six panels are due to decon cycles. These results are derived using the "stretch only" fitting. The "stretch/sharpen" and hybrid symmetric fits give essentially the same wavelength calibration results.

## 4  Spectrally resolved ILS calibration

The ILS functions are derived for each spectral window, spatial footprint, band, and day when OCO-2 made solar measurements. Solar observing orbits within a day are averaged to reduce computation cost. As an example, Fig. 5 shows the spectrally resolved fitting results of ILS functions using daily averaged solar spectra at footprint 6 on 29 March 2015. The three columns of Fig. 5 represent O2A, WCO2, and SCO2 bands, respectively. The first row (Fig. 5a–c) displays the FWHM of the ILS fitted using five different functions, defined in Section 2.2, and at each fitting window (the fitting windows are defined in Fig. 2). The 2016 version of solar line list is used in these fits. The fitted FWHM generally follows the FWHM of preflight ILS well. The "stretch only" ILS function shows the best agreement with preflight ILS, because only a stretch term is fitted, and the structure of preflight ILS is fully preserved. The Gaussian asymmetric function underestimates ILS FWHM due to shape mismatch (see Fig. 6), but successfully captures the spectral variation of FWHM. The second row (Fig. 5d–f) shows the fitting residuals for each band using different ILS functional forms and the root mean squares (RMS) of concatenated residuals from different fitting windows. In most cases, the "stretch/sharpen" fitting shows the lowest residual RMS, followed by the "stretch only" fitting. The hybrid asymmetric/symmetric functions show better fitting residuals in the WCO2 band, where the preflight ILS are very close to a flat-top Gaussian. In the third row (Fig. 5g–i), the preflight ILS functions are directly convolved with solar reference spectra generated from different versions of solar line lists, and only the polynomial scale factors are fitted (no ILS fitting). Hence the residuals of fitting using the 2016 line list (black lines in Fig. 5g–i) are comparable to the residuals in the



second row, Fig. 5d–f. The RMS given by fitting ILS functions are generally smaller than those obtained using just the preflight ILS, indicating that the fitted ILS can better represent the observed solar spectra. The solar line list was significantly improved from the 2013 to 2015 version in the O2A band, where some missing/inaccurate solar lines were corrected (Fig. 5g). The 2016 version further improves the O2A band and is identical to the 2015 version in the WCO2 and SCO2 bands. Hence the following

5    analyses use the 2016 solar line list.

Figure 6 compares the fitted ILS using different function forms with the preflight ILS at three wavelengths for each band. The three columns represent the three OCO-2 bands, and three fitting windows from each band are selected. The same data from Fig. 5 are used, so the FWHM of the fitted ILS can be found in Fig. 5a–c. Generally, the hybrid Gaussian functions capture the preflight ILS in the WCO2 band and the high wavelength end of the O2A band. The Gaussian asymmetric function

10   better represents the SCO2 band and the low wavelength end of O2A band. This partially explains the inconsistent FWHM using hybrid functions at the first fitting window in the O2A band (Fig. 5a) and the good match of FWHM using Gaussian asymmetric function at the last fitting window in the SCO2 band (Fig. 5c). The ILS fits are essentially symmetric even when asymmetric term(s) can be adjusted (the hybrid and Gaussian asymmetric functions), also agreeing with the preflight ILS. The "stretch/sharpen" and "stretch only" fitting results are very similar to and essentially overlap with the preflight ILS in Fig. 6.

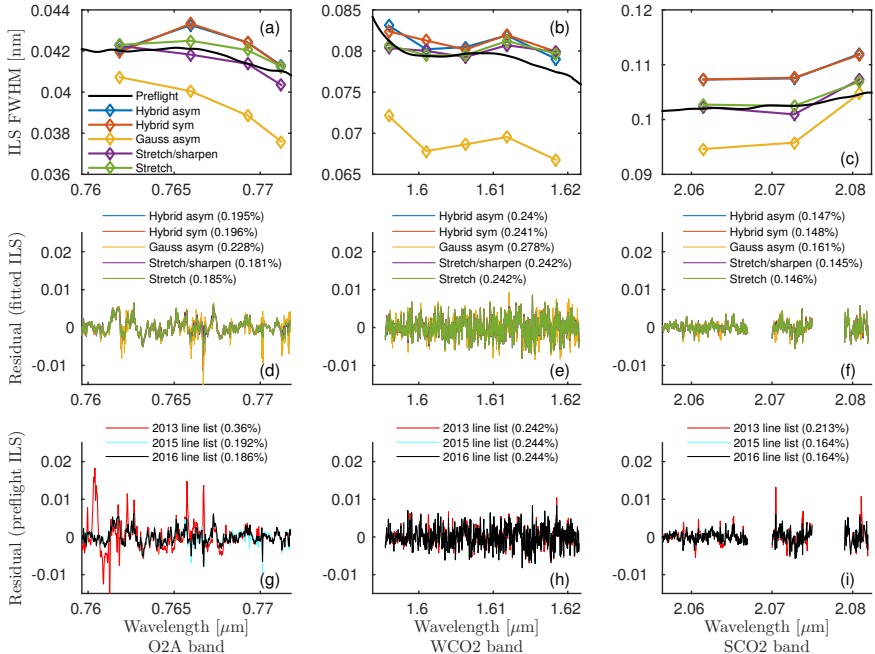

**Figure 5.** (a–c) FWHM of ILS derived using five different ILS functions at each fitting window for three OCO-2 bands. The black line denotes preflight ILS FWHM at each spectral pixel. (d–f) Fitting residuals at each window using different ILS function forms and the RMS of concatenated residuals from all fitting windows. (g–i) Residuals from fitting only the preflight ILS convolved with different versions of solar line lists and polynomial scale factors. This analysis is for footprint 6 on 29 March 2015.



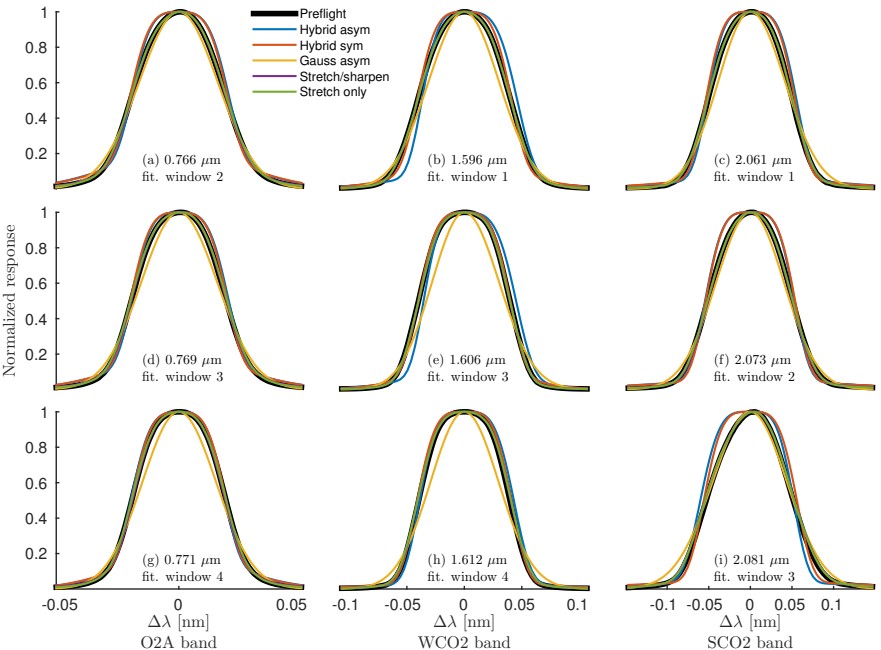

**Figure 6.** Comparison of the derived ILS functions using different function forms with the preflight ILS at three fitting windows for each band. The window numbers are consistent with Fig. 2. The same data from Fig. 5 are shown (footprint 6 on 29 March 2015).

## 5 The impact of spectral sampling on derived ILS using analytical functions

Figure 7 shows the time series of ILS FWHM at daily resolution derived using hybrid asymmetric functions for all spectral windows and all footprints in the O2A band. The first fitting window (0.76–0.764 $\mu$m) shows the most significant temporal variation, with the ILS FWHM changing by $> 3$ %. This change also seems to be driven by a long period forcing other than

the decon cycle, and the changing pattern for each footprint is very different. However, these temporal patterns are completely invisible in the "stretch only" fitting (Fig. 8). Among the five fitting functions, only the functional forms containing a flat-top Gaussian (hybrid asymmetric and symmetric Gaussian) show these features. Hence these are unlikely to be real changes of the OCO-2 ILS. The derived hybrid Gaussian FWHM show strong correlations with the wavelength shift, but the relationships are drastically different for each footprint (Fig. 9), implying that the potential biases of derived FWHM may be related to the

positioning of spectral sampling.

The OCO-2 detectors are slightly tilted with respect to the slit orientation. The largest tilts are seen in the SCO2 band, followed by the O2A, with a much smaller tilt in the WCO2 band (Frankenberg et al., 2015). As a result, the spectral sampling grids are strongly footprint-dependent. To test the ideal impact of spectral sampling positions on the derived ILS FWHM, a highly oversampled solar spectrum was constructed by convolving the high resolution solar reference spectrum and OCO-2

preflight ILS (Fig. 10a). This solar spectrum was then sampled at OCO-2 spectral grids (with wavelength calibration applied) to simulate the solar spectra observed by each footprint.





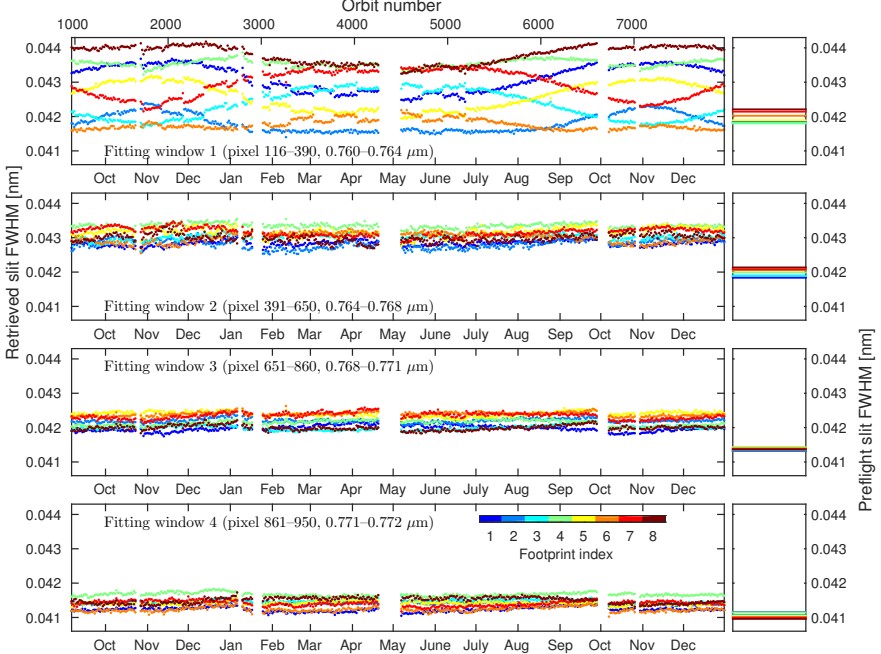

**Figure 7.** Left column: temporal variation of derived ILS FWHM using hybrid asymmetric Gaussian function in four fitting windows for all spatial footprints at the O2A band. Right column: average preflight ILS FWHM in those fitting windows.

In the O2A band, the spectral sampling interval decreases with wavelength due to the grating anamorphic magnification, leading to increasing numbers of samples per FWHM from 2.5 to 3.4 (Fig. 10a). Figure 10b-e zoom in at the peaks of two strong solar lines near 762 and 770 nm (labeled separately by red squares in Fig. 10a) and show the spectral sampling grids of the eight OCO-2 footprints. Because of the annually varying wavelength shifts (Fig. 4), these spectral sampling locations

5  are changing on different days of year with a maximum shift of 0.004 nm, or $1/4$ of the spectral sampling interval (comparing Fig. 10b with Fig. 10d and Fig. 10c with Fig. 10e). Since the spectral sampling is denser near 770 nm (Fig. 10c and e) than 762 nm (Fig. 10b and d), it better captures the solar line shape.

In Fig. 11, the modeled, highly oversampled solar spectrum is resampled at a sliding spectral grid, starting at the spectral sampling grid of footprint 1. Since the O2A band spectral sampling intervals are $< 0.016$ nm (Fig. 10a), sliding the spectral

10  sampling grid by 0.016 nm covers all possible spectral sampling grids for all footprints, assuming the spectral sampling intervals are similar for different footprints (a good assumption for narrow spectral windows used here). The ILS FWHM are then derived using the same ILS fitting methods at each sliding increment. In addition to the five ILS function forms used previously, a "Super Gaussian" is also tested as proposed recently by Beirle et al. (2016):

$$S(\Delta\lambda) = A\exp\left(-\left|\frac{\Delta\lambda}{h_{sg}}\right|^{k}\right),$$

(3)





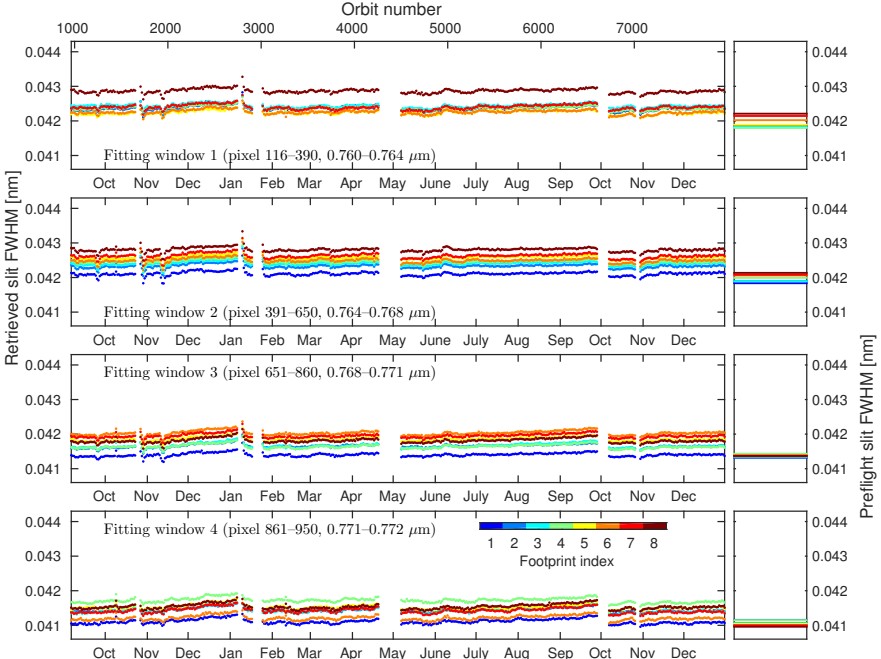

**Figure 8.** Similar to Fig. 7, but the ILS was fitted by stretching the preflight ILS.

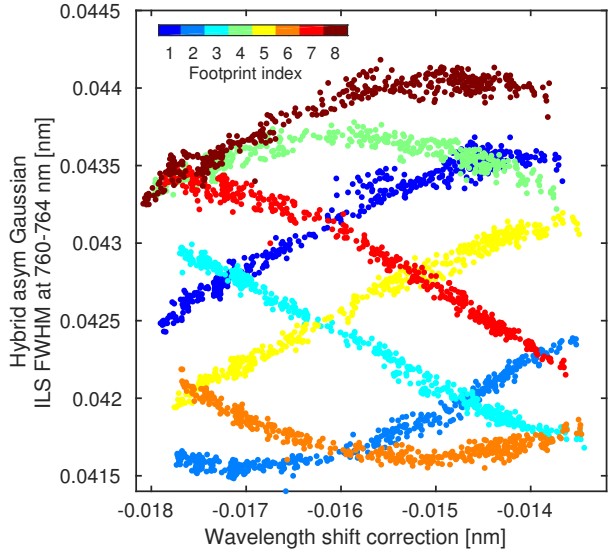

**Figure 9.** Correlation between the derived ILS FWHM using hybrid asymmetric function and the wavelength shift of each footprint.





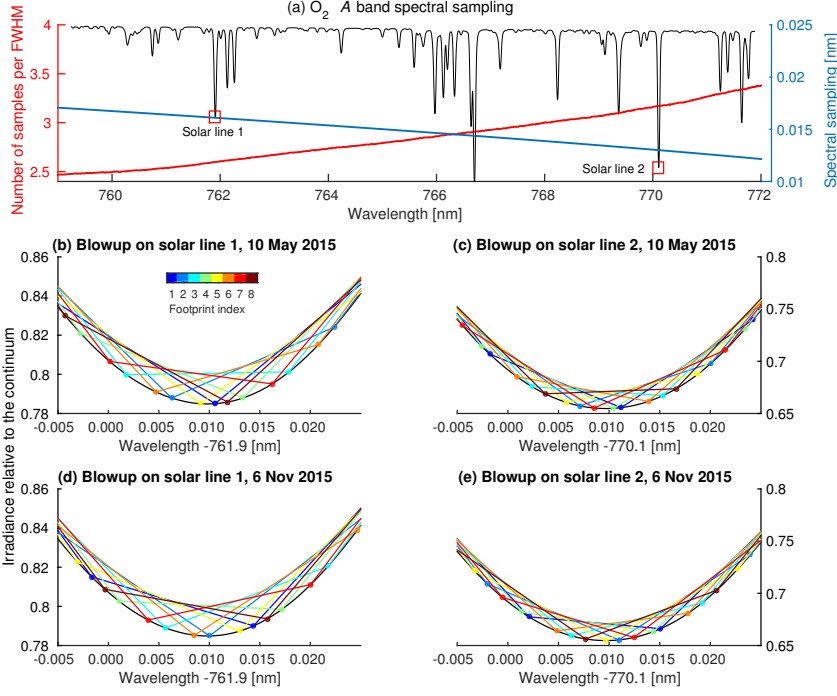

**Figure 10.** (a) Spectral sampling intervals and number of spectral samples per ILS FWHM in the O2A band. A modeled solar spectrum using preflight ILS is shown as the black line. (b and c) Spectral sampling grids by eight footprints on 10 May 2015 near the top of two solar lines marked in (a) by red squares. (d and e) Similar to (b and c) but show the spectral sampling grids on 6 November 2015.

where $A$ is a normalization factor, $h_{sg}$ is the half-width at $1/e$ of the maximum, and $k$ is a shape factor ($k = 2$ gives standard Gaussian; $k = 4$ gives flat-top Gaussian; see Fig. 3b for their shapes).

Because all the sampling cases are based on the same spectrum, the derived ILS should be the same. However, the derived ILS FWHM using hybrid Gaussian functions and the Super Gaussian show remarkable periodic biases at the 761–763 nm window (Fig. 11a) compared to the "true" FWHM derived by fitting the original, oversampled solar spectrum (Fig. 11b). Similar periodic bias is visible in the asymmetric Gaussian fitting, but much smaller ($< 0.2\%$ compared to $5\%$ in the hybrid and Super Gaussian cases). The result from the hybrid asymmetric Gaussian has a phase shift relative to the hybrid symmetric and Super Gaussian, indicating that the asymmetry terms may further complicate the dependence of the derived ILS functions on sampling grid position. In contrast, the impact of sampling grid position is negligible for the "stretch/sharpen" and "stretch only" fittings. These periodical biases are also much smaller for all ILS function forms at 770–772 nm (Fig. 11c–d), where number of samples per FWHM is significantly higher (3.3 vs. 2.6).

Figure 11 demonstrates that biases in the fitted ILS FWHM can be introduced by the positioning of spectral sampling grid. This is because when the sampling is inadequate, the contribution of the flat-top Gaussian part of hybrid Gaussian functions is very sensitive to the positioning of sampling points at the peak region of solar lines (similar for Super Gaussian, which can





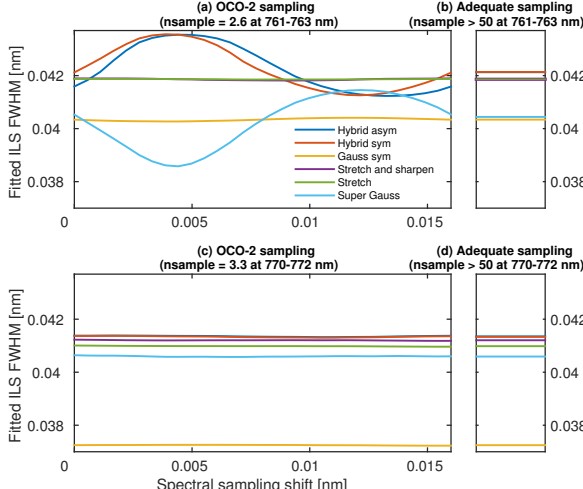

**Figure 11.** (a) The modeled, highly oversampled solar spectrum at 761–763 nm was sampled using a sliding grid. The sampling grid of footprint 1 was used as the starting point. The sampled solar spectra was then fitted using six different ILS functions. (b) Fitting results using the oversampled solar spectrum directly. (c and d) Similar to a and b but at 770–772 nm.

transform continuously between flat-top and standard Gaussians by just varying $k$). OCO-2 is working right on the edge of resolving the high frequency structure of the solar and atmospheric lines. That makes it extremely sensitive to the exact shape of the edge of the ILS. If OCO-2 were at significantly lower or higher spectral resolution, then these ILS edges would not be "beating" against the observed lines so much. This is probably the reason why similar biases were not found in the ILS of
existing space- and air-borne spectrometers even with similar number of spectral samples per FWHM. If there are sufficiently dense spectral samples, the biases can also be mitigated. For example, sampling the modeled solar spectrum at 3 samples per FWHM (instead of 2.6 by the OCO-2 grid) for 761–763 nm reduces the bias by 10 times for the hybrid Gaussian ILS. Therefore this bias is not as significant in the other windows and the other bands, where the sampling is denser.

     Another implication of Fig. 11 is that the ILS FWHM derived from different function forms and the preflight ILS FWHM
are not directly comparable, because even when fitting the same oversampled spectra, these methods give different results (Fig. 11b and d). The fitting methods that preserve the structures of preflight ILS (i.e., the "stretch only" and "stretch/sharpen" fittings) are more representative of the true FWHM.

     Figure 12 shows the range of spectral sampling grid shifts of the eight OCO-2 footprints and their theoretical impact on the derived hybrid asymmetric Gaussian function FWHM. Compared to Fig. 9, the ideal FWHM responses in Fig. 12 closely
agree with correlations between the actual derived ILS FWHM and wavelength shifts. This explains the variation patterns in Fig. 7 and confirms that these are artifacts induced by the relative positioning of spectral sampling grids. Therefore, the "stretch/sharpen" and "stretch only" fittings better represent the possible variations of ILS in reality than analytical functions.





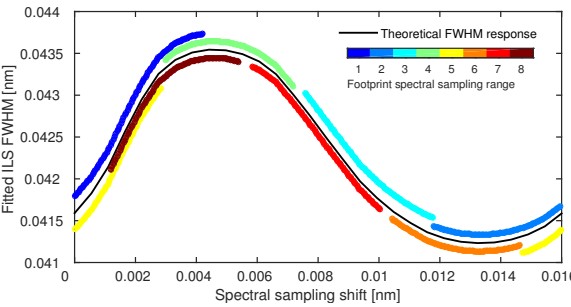

**Figure 12.** The black line shows the derived ILS FWHM using the hybrid Gaussian function at sliding OCO-2 spectral sampling grid (same as the "Hybrid asym" line in Fig. 11a). The color lines show the range of the wavelength Doppler shift of the spectral sampling grid for each footprint, relative to the spectral sampling grid of footprint 1.

## 6   Temporal variation of OCO-2 on-orbit ILS

The temporal variations of derived ILS FWHM for all footprints in the O2A band have been shown in Fig. 7 (hybrid asymmetric fitting) and Fig. 8 ("stretch only" fitting). Although the first fitting window results for the hybrid Gaussian functions are biased due to inadequate sampling, the results for the rest of the fitting windows show close agreement between the two forms of ILS functions. Both methods show that the inter-footprint differences are larger for on-orbit ILS FWHM than the preflight
ILS. The ILS FWHM of footprint 8 in the first fitting window is significantly larger than the other footprints, which is even visible in the biased hybrid Gaussian fitting (also see Fig. 9 where the curve for footprint 8 is higher than the others). Changing patterns are similar for all footprints and fitting windows; the ILS FWHM dropped slightly after each decon and then increased almost linearly. Therefore, it is possible to derive an ILS for the whole band. The varying biases of the hybrid Gaussian ILS
seen for the first fitting window are still present when fitting the entire O2A band. However, according to Fig. 9 and Fig. 12, the spectral sampling grid of footprint 4 is relatively insensitive to this bias. Figure 13a–b show the O2A band ILS FWHM derived by fitting the five ILS functional forms to footprint 4, using daily averaged regular solar spectra and solar Doppler spectra, respectively. All FWHM values are normalized to their median values. The FWHM from all ILS functional forms except for "stretch/sharpen", which is much less variant, agree well with each other and gradually increase between decon
events. Figure 13c displays the sharpen term of "stretch/sharpen" fitting. A smaller value of this term indicates wider wings of ILS, as shown by Fig. 3d. Because the "stretch/sharpen" fitting fully decouples the homogeneous stretch/squeeze of the ILS and broadening of ILS wings, the fact that only the sharpen term responds to the decon cycles implies that the apparent ILS widening captured by the other fitting methods is mainly caused by widening of the wings. This can also be captured by hybrid Gaussian fittings, where the flat-top Gaussian mainly represents the central part of ILS, and standard Gaussian captures
the wings (see Fig. 3b). Figure 13d shows the widths of the standard Gaussian fitted by the hybrid Gaussian functions. The widths of the standard Gaussian components in the hybrid Gaussian functions follow a similar trend to the FWHM. Between the decon events in May and September 2015, the widths of the standard Gaussian components increased by $\sim 5\ \%$, whereas





the widths of flat-top Gaussian components varied by less than 1 % (not shown). This supports the conclusion that the apparent broadening of the ILS functions is driven by broadening of the wings instead of a homogeneous stretch of the entire ILS. As such, the "stretch/sharpen" fitting appears to be the best way to capture the on-orbit behavior of OCO-2 ILS.

The physical cause of the broadening of ILS wings is believed to be the accumulation of a very thin layer of ice on the antireflective (AR) coating of the FPAs, which is also the cause of the fast degradation of O2A FPA sensitivity between decons (see Crisp et al. (2016) for a more detailed discussion). The ice layer enhanced the reflectance of the FPA, and the reflected light might be scattered back to the FPA by the other optical components. Because the scattered light hitting a given pixel is likely reflected by a broad range of pixels, this effect can be quantified as widening of the ILS wings, i.e., the "sharpen" term. There are no significant temporal trends observed in the derived ILS of the WCO2 and SCO2 bands, consistent with the fact that the AR coatings of the WCO2 and SCO2 FPAs are insensitive to ice accumulation and hence much less light is reflected (Crisp et al., 2016).

Another way to quantify the broadening of the ILS wings is by fitting an additive offset term that simulates the spatial distribution of the scattered light on the FPA. We found that fitting an additive offset and the "stretch/sharpen" fitting gives very similar results, using synthetic solar spectra with realistic OCO-2 ILS, SNR, and varying additive offsets. In this study, we fit the ILS, instead of the offset, because the ILS fitting may reveal other changes in the ILS shape and the offset can be successfully fit away by ILS, as shown in the following section (Fig. 14).

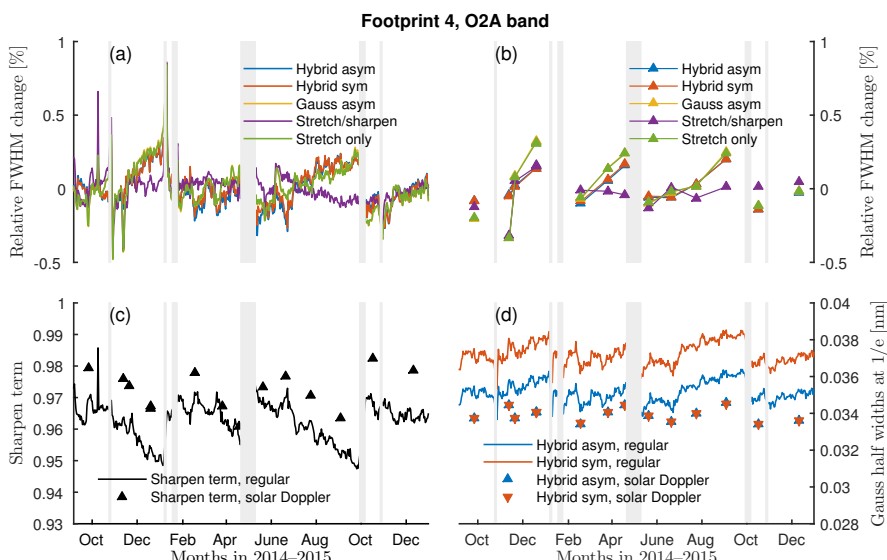

**Figure 13.** (a) Relative variations of derived ILS FWHM from regular solar spectra using five fitting methods at footprint 4 of O2A band. The FWHM are normalized by their medians and subtracted by unity. (b) similar to (a) but using solar Doppler spectra. (c) sharpening term in the "stretch/sharpen" fitting, using both regular and solar Doppler spectra. (d) Half widths at $1/e$ derived using hybrid asymmetric/symmetric function fitting, using both regular and solar Doppler spectra. The gray bands show the decon events.





## 7 Verifying solar-derived ILS with Earthshine spectra

As noted previously, the transmissive solar diffuser does not uniformly fill the aperture of OCO-2, which may induce ILS artifacts specific to the solar spectra. Therefore, it is necessary to confirm that the temporal trends found in the previous section are also present in the Earthshine spectra. The OCO-2 retrieval of solar-induced fluorescence (SIF) provides a straightforward

way to test this in the O2A band. The fluorescence signals are retrieved as a relative offset from the Earthshine spectra using two microwindows encompassing solar lines near 758 and 770 nm (Frankenberg et al., 2011). However, the retrieved offset is a combination of chlorophyll fluorescence and instrumental artifacts (e.g., the unaccounted changes of ILS or additive offsets introduced by the instrument). The retrieved fluorescence signals then need to be corrected based on retrievals in barren areas where no chlorophyll fluorescence is expected (deserts, ocean areas with negligible productivity). Figure 14 shows the

uncorrected "SIF" signals retrieved in the Earth's barren surfaces (black lines). For the official OCO-2 fluorescence product, this time-varying bias is subtracted from the overall offset fit to differentiate between fluorescence and instrument-related biases.

Here, we used the same fluorescence retrieval principle applied to the solar spectra, where no confounding factors of Earthshine spectra (varying albedo, scattering by aerosols/clouds) are present but with the solar diffuser. The results (blue lines)

show very similar variations compared to the spurious signals in the uncorrected SIF retrievals from the Earth-shine data. To test if this can be explained by ILS change, the derived ILS using the "stretch/sharpen" method are applied instead of the pre-flight ILS in the fluorescence retrieval. The trends in the retrieved offset are no longer significant (red lines). This suggests that (1) the widening of the ILS wings in the O2A band is a real feature, not introduced by the solar diffuser, and (2) the spurious fluorescence signals seen in barren areas on the Earth may be mitigated by taking the temporal variation of ILS into account.

Currently, the time-dependent fluorescence bias is corrected in a post-processing step. The findings here also show that the widening of the ILS is indeed closely related to an additive offset term, which is the very same effect as the fluorescence term from chlorophyll or the a contribution from scattered light by the buildup of an ice layer on the detector. We found that for every $+1\%$ offset relative to the continuum level, the sharpen term (indicating widening of ILS wings) decreases by $\sim 5\%$. For the real data, the sharpen term decreases by $\sim 2.5\%$ between decon events (Fig. 13c), corresponding to a $\sim 0.5\%$ additive

offset. This agrees well with the changes in the additive offsets derived from both solar and Earthshine spectra.

## 8 Conclusions

This study presents the post-launch, on-orbit characterization of the ILS of the OCO-2 instrument. Different functional forms of the ILS are fitted to match a high resolution solar reference spectrum to the OCO-2 solar observations. The 2016 version of TCCON solar line list shows improvement over the current solar model used in the v7 L2 retrieval algorithm, mainly in

the O2A band. The analytical functions used in the previous space-borne grating spectrometers are found to be inadequate to characterize the OCO-2 ILS. The asymmetric Gaussian function captures the spectral variation of the ILS FWHM, but gives the largest fitting residual and cannot fully represent the top-hat shape of OCO-2 ILS. At the OCO-2 spectral resolution, the hybrid asymmetric Gaussian function that are currently used in suborbital spectrometers (ACAM (Liu et al., 2015), Geo-



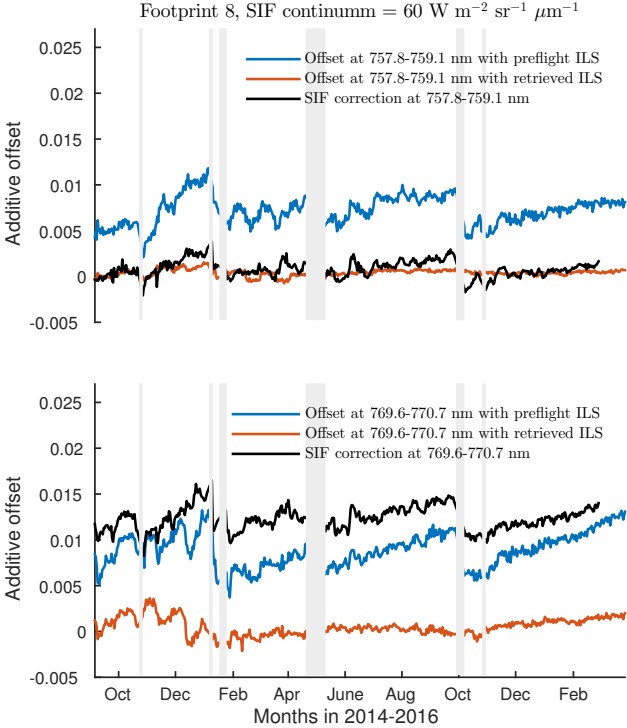

**Figure 14.** Retrieved offset from the Earthshine spectra as fraction of continuum level in barren areas using preflight ILS (black); retrieved offset from solar spectra using preflight ILS (blue); and retrieved offset using solar-derived ILS through the "stretch/sharpen" fitting (red). The top and bottom panels show the SIF retrieval windows near 758 nm and 770 nm, respectively. The gray bands show decon events.

TASO (Nowlan et al., 2016)) may introduce spurious variations depending on the spectral sampling position when ILS FWHM is undersampled. The newly proposed Super Gaussian function also has the similar issue. The empirical ILS functional forms that preserve the detailed structure of preflight ILS are insensitive to the sampling position and hence preferable to the analytical forms in the case of OCO-2.

5    An "apparent" widening of the ILS, by up to $0.5$ %, is found between decon events in the O2A band, driven by broadening of only the wings of ILS. Therefore, the fitting function that fully decouples the homogeneous stretching and widening of the ILS wings (the "stretch/sharpen" fitting) is the one to capture the on-orbit behavior of the OCO-2 ILS. The broadening of the ILS wings is also supported by the SIF retrievals, where spurious SIF signals in barren areas on the Earth can be mitigated by applying the time-variant ILS derived from solar spectra. This confirms the effectiveness of using daily solar measurements to

10   monitor the on-orbit changes of ILS. To account for the scattered light in O2A band, it is also possible to adjust an additive offset that simulates the spatial distribution of the scattered light on the FPA. Ultimately, these corrections will have to be tested on $X_{CO_2}$ retrievals, which is ongoing research for the v8 L2 algorithm.

Compared to the existing space-borne grating spectrometers, the spectral resolving power and retrieval accuracy requirement of OCO-2 are unprecedented. In addition, the rotational-vibrational bands resolved by OCO-2 have finer features than



absorption in the UV/visible. These help explain why the established methods for the UV/visible satellites do not suffice to characterize OCO-2 ILS. This study also has implications for future missions targeting at high resolution, high accuracy greenhouse gas retrievals (TROPOMI, OCO-3, and Sentinel-5), where accurate knowledge of the fine structure of ILS and the on-orbit variation may be critical.

5  *Acknowledgements.* This study is supported by the NASA OCO-2 science team (NNX12AH38G) and the Smithsonian Institution. We acknowledge Geoffrey Toon at JPL for making the high resolution solar spectra generated from newer versions of solar line list available. We also thank Gonzalo Gonzalez Abad at SAO, Debra Wunch at Caltech, Steffen Beirle at MPI-C, and James McDuffie, Lars Chapsky, Gregory Osterman, and Brendan Fisher at JPL for helpful discussions. Part of the research described in this paper was carried out at the Jet Propulsion Laboratory, California Institute of Technology, under a contract with the National Aeronautics and Space Administration.



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
