# Peer review of "Characterization of the OCO-2 instrument line shape functions using on-orbit solar measurements"

_Atmospheric Measurement Techniques, 2016_

## Referee Comment (RC1) · Anonymous Referee #2 · 11 Jan 2017

The paper analyses the OCO-2 ILS and compares different analytical and empirical parameterizations. The paper is well written, and the findings are of great interest for the retrieval of trace gases from satellite spectra. In particular the dependency of analytical parameterizations on sampling position, as consequence of undersampling of OCO-2, is carved out well. I recommend publication after dealing with the following comments:

1. The introduction is quite vague with respect to previous satellite missions. For instance, to what "existing satellite instruments" (page 1, line 17) do you refer? TOMS? GOME? OMI? GOSAT? Please extend the description of existing instruments relevant for this study, and show the improvement of OCO-2 design in direct comparison to e.g. GOSAT. Also the "species measured by existing satellite instruments" (page 2, line 21) are quite general; please specify, and note that OCO-2 is not the first instrument

measuring CO2.

2. In section 3, various ILS parameterizations are listed which are applied in the following analysis. Later in section 5, the Super Gaussian is introduced and applied as well. Please introduce the Super Gaussian earlier and add it to the list of parameterizations from section 3 on.

3. Page 1, line 12: "induced"

---

## Referee Comment (RC2) · Anonymous Referee #1 · 20 Jan 2017

**General comments**

This study is an accurate characterization of the OCO-2 ILS function from on-orbit solar calibration measurements. The OCO-2 preflight ILS was provided in tables because typical line shape functions cannot fit well (Frankenberg et al., 2015). Authors analyzed on-orbit solar spectra fitting by several analytical functions and stretch/sharpen preflight ILS. They showed the ILS fitting result of several solar lines in spectral micro-windows and the temporal variations of ILS FWHMs.

They found the derived O2A ILS sharpen term was changed but the ILS FWHM of O2A band was stable. The sharpen term decreased in time but recovered after decontamination operation. On-orbit ILS is estimated the center is sharper, but the wing is broadened than preflight ILS. They discussed that this O2A ILS wing broadening was related to the reflection from the FPA with ice layer accumulation (Crisp

et al., 2016). The reflection might be affected to SIF retrieval in earth observation spectra. They showed the O2A band had an offset radiance in the SIF retrieval from solar spectra, which have no chlorophyll fluorescence. Temporal variation of on-orbit stretch/sharpen ILS for solar spectra can explain the SIF offset. They concluded an accurate knowledge of the ILS on-orbit variation might be useful for precise $XCO_2$ retrieval.

Overall, this study showed the OCO-2 ILS on-orbit assessment method. This result is very useful for precise retrieval in long-term observation. In future study, $XCO_2$ is expected to become much better accuracy with reducing spectral residual by applying the long-term ILS change. This paper is well written with sufficient detailed explanation in the scope of AMT. I recommend publication with some revisions for further clear explanation.

**Specific comments**

I have some comments and recommendations from Section 2 to Section 7.

**1. Introduction**

Authors briefly address the OCO-2 mission introduction and ILS calibration importance for precise $XCO_2$ retrieval. They also introduce previous studies on on-orbit ILS investigations by other atmospheric spectroscopic missions.

**2. Instrument and data analysis**

In this section, authors show the OCO-2 sensor specification, on-orbit solar calibration, ILS fitting analysis method focused on Fraunhofer lines by using several analytical functions and modified preflight ILS. They briefly explain how the preflight ILS table was constructed in previous studies.

In the following Section 6 (Page 16, L 2- 3): As such, the "stretch/sharpen" fitting appears to be the best way to capture the on-orbit behavior of OCO-2 ILS. The "stretch/sharpen" fitting is important in this paper result as shown in the above sentence. However, in Section 2.2 (Page 6, L19-21) the definition appears only in sentence explanation. I recommend you show the equation in Section 2.2.

**3. Wavelength calibration of solar spectra**
In this section, authors show the temporal variation of the wavelength shift and the squeeze term from solar spectra of O2A, WCO2 and SCO2 bands.

Is your study comparable to the wavelength shift and squeeze term of Crisp et al. (2016) result? You can compare with peak-to-peak variations of the shift and the squeeze. Derived consistency or difference between Earth radiance spectra and Solar radiance spectra is an important information.

**4. Spectrally resolved ILS calibration**
In this section, authors show FWHMs and spectral residuals using several analytical ILS functions derived from solar spectra. Analyzed parameters are compared with the preflight ILS at each wavelength window.

L10-12: This sentence is not correct.
L11: at the first fitting window in the O2A band (Fig. 5a) -> at the fitting window 2 in the O2A band (Fig. 5a and Fig. 6a). Correct the corresponding figure number for the inconsistent FWHM.
L12: at the last fitting window in the SCO2 band (Fig. 5c) -> at the fitting window 3 in the SCO2 band (Fig. 5c and Fig. 6i). Correct the corresponding figure number for the good match FWHM.

[Figure]

L13-14: The "stretch/sharpen" and "stretch only" fitting results are very similar to and essentially overlap with the preflight ILS in Fig. 6.

If the sharpen term < 1 appears in Fig. 6, I recommend you mention wing broadening briefly after the last sentence in Section 4 to become consistent with the following Section 6.

**5. The impact of spectral sampling on derived ILS using analytical functions**

In this section, authors make a sensitivity test for the ILS estimation to the spectral sampling.

L 3: The first fitting window (0.76-0.764 $\mu$m) -> The fitting window 1. Correct as a definition.

By stretching ILS analysis, temporal variations of wavelength shift (Fig. 4) and ILS FWHM (Fig. 8) for each footprint are the same in order. However, by hybrid asymmetric Gaussian ILS analysis, temporal variation of FWHM for each footprint is not in order especially in the fitting window 1. One candidate cause is the ILS FWHM strongly relates to the wavelength shift in the fitting window 1 as shown in Fig. 9. How is an example of the other windows like Fig. 9?

In Fig. 9, the variation of ILS FWHM of asymmetric Gaussian is 0.001nm (0.0425 ~ 0.0435nm), and the corresponding variation of wavelength shift is 0.004nm (-0.018 ~ -0.014nm). However, Fig. 11a simulation (in blue) shows slightly larger ILS FWHM variation as 0.002nm (0.042 ~ 0.044nm) corresponding to wavelength shift 0.004nm (0 ~ 0.004nm). Why are they different?

**6. Temporal variation of OCO-2 on-orbit ILS**

In this section, authors show temporal variation of OCO-2 on-orbit ILS features in FWHM, sharpen term, and wing width. They discuss that this O2A ILS wing broadening up to 0.5 % between decontaminations is related to the reflection from the FPA with ice layer accumulation.

L 15-16: A smaller value of this term -> A smaller value (less than 1) of this term. To become clear.

Figures 13c and 13d show that monthly solar Doppler spectra is slightly sharper and narrower than daily regular solar spectra. Why do the sharpen term and the wing width have differences between the two solar calibrations?

Figure 3b shows the derived hybrid Gaussian is composed of flat-top Gaussian (ILS center) and standard Gaussian (ILS wings). This figure looks the ILS width at 1/e maximum comes almost from the flat-top Gaussian part. Is it consistent with Fig. 13d?

**7. Verifying solar-derived ILS with Earthshine spectra**

In this section, authors retrieve SIF and radiance correction offset from solar spectra, which have no fluorescence. They compare radiance correction offset of earth's barren surface spectra, solar spectra using preflight ILS and solar spectra using "stretch/sharpen" preflight ILS.

Figure 14 shows the SIF correction of earth radiance (black) from 758nm line is almost zero, but 770nm has an offset. However, the SIF corrections of solar radiance (both blue and red) are almost the same offset at both wavelengths. What is the cause candidate? It might be corresponding to the spectral residual (Fig. 5g).

Is the SIF retrieved stretch/sharpen term in Fig. 14 the same as Fig. 13?

**8. Conclusions**

Authors summarize this paper results of OCO-2 on-orbit ILS analysis by using solar spectral calibration. They propose this ILS corrections are tested on $XCO_2$ retrieval for L2 v8 algorithm for improvement.

**Technical corrections**

L 4: In Eq. (1), $\lambda$ definition does not appear.

L 6: ILS functions used in this study first appear as a list. So, "listed above" -> "listed below". Or revise/add as a list in Section 2.2.

L 5- 6: "the FWHM of the ILS fitted using five different functions, defined in Section 2.2"
Five analytical functions are addressed and defined in Section 2.2, but apparently listed in Section 3.

Figure 7 caption: at the O2A band -> in the O2A band.

[Figure]

---

## Author Comment (AC1) · 14 Feb 2017

Response to Referee #2:

We appreciate the very helpful feedback from the referee. The referee's comments are listed in *italics*, followed by our response in blue. New/modified text in the manuscript is in **bold**.

*1. The introduction is quite vague with respect to previous satellite missions. For instance, to what "existing satellite instruments" (page 1, line 17) do you refer? TOMS? GOME? OMI? GOSAT? Please extend the description of existing instruments relevant for this study, and show the improvement of OCO-2 design in direct comparison to e.g. GOSAT.*

Following the referee's suggestion, previous satellite missions that measure $CO_2$ column abundance are included and compared with OCO-2. The sentences at page 1, line 16 are modified to:

"**To achieve its mission goal, OCO-2 was designed to measure the reflected sunlight in near infrared $O_2$ and $CO_2$ bands with significantly higher sensitivity, spectral and spatial resolution, and spatial coverage requirements than previous satellite $CO_2$ measurements. For example, the nadir resolution of OCO-2 is less than $1.3 \times 2.3$ km$^2$, much finer than those of SCIAMACHY ($30 \times 60$ km$^2$) and GOSAT/TANSO-FTS (diameter of 10.5 km). The OCO-2 instrument aims to measure the column-averaged $CO_2$ dry air mole fraction, $XCO_2$, with uncertainties near 1 ppmv (0.25 % of current $XCO_2$) on regional-to-continental scales (Crisp et al., 2004; Crisp, 2008; Frankenberg et al., 2015), also significantly smaller than what achieved by SCIAMACHY and GOSAT (Buchwitz et al., 2005, Butz et al., 2011).**"

*Also the "species measured by existing satellite instruments" (page 2, line 21) are quite general; please specify,*

This sentence was modified to:

"**The retrieval accuracy requirement for OCO-2 (~0.25% for XCO2) is also much higher than those for the species measured in the UV/visible range (e.g., the required precision for air quality species is usually >10% (Zoogman et al., 2017)), so small ILS differences that may be tolerated in other instruments may jeopardize the XCO2 retrieval.**"

*and note that OCO-2 is not the first instrument measuring CO2.*

Two previous missions (SCIAMACHY and GOSAT) that measure column $CO_2$ have been added.

*2. In section 3, various ILS parameterizations are listed which are applied in the following analysis. Later in section 5, the Super Gaussian is introduced and applied as well. Please introduce the Super Gaussian earlier and add it to the list of parameterizations from section 3 on.*

Revised as suggested. The Super Gaussian is now defined in Section 2.2 with the other ILS function forms and included in the discussion from section 3 on.

*3. Page 1, line 12: "induced"*

Revised.

---

## Author Comment (AC2) · 14 Feb 2017

Response to Referee #1:

We appreciate the very helpful feedback from the referee. The referee's comments are listed in *italics*, followed by our response in blue. New/modified text in the manuscript is in **bold**.

*2. Instrument and data analysis*

*In this section, authors show the OCO-2 sensor specification, on-orbit solar calibration, ILS fitting analysis method focused on Fraunhofer lines by using several analytical functions and modified preflight ILS. They briefly explain how the preflight ILS table was constructed in previous studies.*

*In the following Section 6 (Page 16, L 2- 3): As such, the "stretch/sharpen" fitting appears to be the best way to capture the on-orbit behavior of OCO-2 ILS.*

*The "stretch/sharpen" fitting is important in this paper result as shown in the above sentence. However, in Section 2.2 (Page 6, L19-21) the definition appears only in sentence explanation. I recommend you show the equation in Section 2.2.*

Equation added as suggested.

*3. Wavelength calibration of solar spectra*

*In this section, authors show the temporal variation of the wavelength shift and the squeeze term from solar spectra of O2A, WCO2 and SCO2 bands.*

*Page 7*

*Is your study comparable to the wavelength shift and squeeze term of Crisp et al. (2016) result? You can compare with peak-to-peak variations of the shift and the squeeze. Derived consistency or difference between Earth radiance spectra and Solar radiance spectra is an important information.*

The peak-to-peak variations of the shift and squeeze terms in this work derived from Solar data did show strong similarity with the terms derived from Earth data in Crisp et al. AMTD (2016, now published in AMT 2017). The following sentences have been added in the manuscript (Page 7, Line 31):

"**The similar effects on the dispersion shift/squeeze terms have been consistently observed in the L2 full physics retrievals using the Earthshine spectra (Figure 7 in Crisp et al. (2017) shows Earthshine data from one footprint). The peak-to-peak variations of the shift/squeeze terms (after removing the Doppler shifts in the shift term) agree closely between solar and Earthshine spectra.**"

*4. Spectrally resolved ILS calibration*

*In this section, authors show FWHMs and spectral residuals using several analytical ILS functions derived from solar spectra. Analyzed parameters are compared with the preflight ILS at each wavelength window.*

*Page 9*

*L10-12: This sentence is not correct.*

*L11: at the first fitting window in the O2A band (Fig. 5a) -> at the fitting window 2 in the O2A band (Fig. 5a and Fig. 6a). Correct the corresponding figure number for the inconsistent FWHM.*

Revised as suggested.

*L12: at the last fitting window in the SCO2 band (Fig. 5c) -> at the fitting window 3 in the SCO2 band (Fig. 5c and Fig. 6i). Correct the corresponding figure number for the good match FWHM.*

Revised as suggested.

*L13-14: The "stretch/sharpen" and "stretch only" fitting results are very similar to and essentially overlap with the preflight ILS in Fig. 6. If the sharpen term < 1 appears in Fig. 6, I recommend you mention wing broadening briefly after the last sentence in Section 4 to become consistent with the following Section 6.*

A sentence is added to the end of Section 4 as suggested:

"**The sharpen terms are slightly less than unity for the O2A band, indicating that the wings of derived ILS are broader than the preflight, and vary with time, as will be shown in Section 6.**"

*5. The impact of spectral sampling on derived ILS using analytical functions*

*In this section, authors make a sensitivity test for the ILS estimation to the spectral sampling.*

*Page 10*

*L 3: The first fitting window (0.76-0.764 µm) -> The fitting window 1. Correct as a definition.*

Done.

*By stretching ILS analysis, temporal variations of wavelength shift (Fig. 4) and ILS FWHM (Fig. 8) for each footprint are the same in order. However, by hybrid asymmetric Gaussian ILS analysis, temporal variation of FWHM for each footprint is not in order especially in the fitting window 1. One candidate cause is the ILS FWHM strongly relates to the wavelength shift in the fitting window 1 as shown in Fig. 9. How is an example of the other windows like Fig. 9?*

For the other fitting windows, the ILS FWHM is not correlated to the wavelength shift due to denser spectral sampling. This was explained in page 14, L7-8 of the AMTD manuscript: "Therefore this bias is not as significant in the other windows and the other bands, where the sampling is denser."

*Page 11*

*In Fig. 9, the variation of ILS FWHM of asymmetric Gaussian is 0.001nm (0.0425 ∼ 0.0435nm), and the corresponding variation of wavelength shift is 0.004nm (-0.018 ∼ -0.014nm). However,*

*Fig. 11a simulation (in blue) shows slightly larger ILS FWHM variation as 0.002nm (0.042 ~ 0.044nm) corresponding to wavelength shift 0.004nm (0 ~ 0.004nm). Why are they different?*

Good catch. They are different because Fig. 11a's simulation used a smaller spectral window (761-763 nm) to zoom in at a group of strong solar lines, whereas Fig. 9 was derived from fitting window 1 that has a wider range (760-764 nm). Wider window (hence more solar lines) smears out some of the artificial ILS FWHM variations. We choose a 2-nm wide spectral window here to make it directly comparable with the other 2-nm wide spectral window at 0.770-0.772 nm, shown in Fig. 11c. To clarify this point, we added the following sentence to Page 14, L16 of the original manuscript:

"**The variations of ILS FWHM in Fig.11a and Fig.12 are slightly larger than in Fig. 9 because Fig. 9 uses a wider spectral range (760-764 nm vs. 761-763 nm) that includes more solar lines and smears out part of the artifacts.**"

*6. Temporal variation of OCO-2 on-orbit ILS*

*In this section, authors show temporal variation of OCO-2 on-orbit ILS features in FWHM, sharpen term, and wing width. They discuss that this O2A ILS wing broadening up to 0.5 % between decontaminations is related to the reflection from the FPA with ice layer accumulation.*

*Page 15*

*L 15-16: A smaller value of this term -> A smaller value (less than 1) of this term. To become clear.*

Revised.

*Figures 13c and 13d show that monthly solar Doppler spectra is slightly sharper and narrower than daily regular solar spectra. Why do the sharpen term and the wing width have differences between the two solar calibrations?*

The most likely explanation is that each daily regular solar spectrum is the trimmed average of ~180 individual frames, which last for about 3 minutes. This introduces an extra Doppler broadening corresponding to the spacecraft-Sun velocity change during these 3 minutes. The solar Doppler spectra are constructed by smaller trimmed-averaging intervals, and the Doppler broadening is not significant. A sentence was added to explain this (Page 16, L1 of the AMTD manuscript):

"**The absolute values are slightly different for regular and solar Doppler data, where the regular solar results are generally broader. This is likely due to more averaging in regular spectra that introduces a constant Doppler broadening of ~1% of the ILS FWHM. However, the trends are consistent for all observations.**"

*Figure 3b shows the derived hybrid Gaussian is composed of flat-top Gaussian (ILS center) and standard Gaussian (ILS wings). This figure looks the ILS width at 1/e maximum comes almost from the flat-top Gaussian part. Is it consistent with Fig. 13d?*

It is true that the ILS width at 1/e (and at half maximum as well) is determined by the width somewhere in the flat-top Gaussian part. However, the narrowing/broadening of the standard Gaussian part can move the flat-top Gaussian part up and down, such that even when the flat-top Gaussian part stays the same, the 1/e (and half maximum) locations of the entire ILS will correspond to different levels of the flat-top Gaussian, and the ILS HWHM/HW1e will be different. One sentence was added to the text to clarify this (Page 16, L1 of the AMTD manuscript):

"**Even when the flat-top Gaussian component stays constant, the ILS HWHM can still be controlled by the standard Gaussian component, which can move the flat-top Gaussian component up and down relatively.**"

*7. Verifying solar-derived ILS with Earthshine spectra*

*In this section, authors retrieve SIF and radiance correction offset from solar spectra, which have no fluorescence. They compare radiance correction offset of earth's barren surface spectra, solar spectra using preflight ILS and solar spectra using "stretch/sharpen" preflight ILS.*

*Figure 14 shows the SIF correction of earth radiance (black) from 758nm line is almost zero, but 770nm has an offset. However, the SIF corrections of solar radiance (both blue and red) are almost the same offset at both wavelengths. What is the cause candidate? It might be corresponding to the spectral residual (Fig. 5g).*

The SIF offset was derived operationally from Earthshine spectra, where the absorption of O2 was taken into account in the 770 nm window. The SIF algorithm also used an older version of the solar line list. The absolute values of the retrieved offset are sensitive to different versions of solar line list, but the trends are consistent.

*Is the SIF retrieved stretch/sharpen term in Fig. 14 the same as Fig. 13?*

The stretch/sharpen terms in Fig. 14 were derived using the SIF microwindows near 758 and 770 nm, while Fig. 13 shows the stretch/sharpen terms derived using the entire band. Therefore the values are slightly different. Text in Section 7 was modified to provide this information (Page 17, L15 of the AMTD manuscript):

"**To test if this can be explained by ILS change, the ILS in the SIF microwindows were first derived using the ``stretch/sharpen'' method, then applied in the fluorescence retrieval, instead of the preflight ILS. The derived stretch/sharpen terms have the same temporal trends as Fig. 13, where the entire band was used, but slightly different values.**"

*Technical corrections*

*Page 5 L 4: In Eq. (1),λ definition does not appear.*

Revised the definition of ILS as "**S(λ) denotes the ILS function that is defined at wavelength grid λ**".

*Page 7 L 6: ILS functions used in this study first appear as a list. So, "listed above" -> "listed below". Or revise/add as a list in Section 2.2.*

Revised as "**listed below**".

*Page 8 L 5- 6: "the FWHM of the ILS fitted using five different functions, defined in Section 2.2"*
*Five analytical functions are addressed and defined in Section 2.2, but apparently listed in*
*Section 3.*

Revised as "**the FWHM of the ILS fitted using six different functions, listed in Section 3**".
Note that the Super Gaussian has been included in the analyses following referee #2' comments,
so six, instead of five, ILS results are shown here.

*Page 11 Figure 7 caption: at the O2A band -> in the O2A band*

Revised.